# Solving conformal defects in 3D conformal field theory using fuzzy sphere regularization

Liangdong Hu [1,2], Yin-Chen He [3] ✉ & W. Zhu [1] ✉

Defects in conformal field theory (CFT) are of significant theoretical and experimental importance. The presence of defects theoretically enriches the structure of the CFT, but at the same time, it makes it more challenging to study, especially in dimensions higher than two. Here, we demonstrate that the recently-developed theoretical scheme, *fuzzy (non-commutative) sphere regularization*, provides a powerful lens through which one can dissect the defect of 3D CFTs in a transparent way. As a notable example, we study the magnetic line defect of 3D Ising CFT and clearly demonstrate that it flows to a conformal defect fixed point. We have identified 6 low-lying defect primary operators, including the displacement operator, and accurately extract their scaling dimensions through the state-operator correspondence. Moreover, we also compute one-point bulk correlators and two-point bulk-defect correlators, which show great agreement with predictions of defect conformal symmetry, and from which we extract various bulk-defect operator product expansion coefficients. Our work demonstrates that the fuzzy sphere offers a powerful tool for exploring the rich physics in 3D defect CFTs.

Defects, as well as their special case–boundaries, are fundamental elements that inevitably exist in nearly all realistic physical systems. Historically, research on defects has played a pivotal role in shaping modern theoretical physics. This includes contributions to the theory of the renormalization group (RG)[1], studies of topological phases[2–4], investigations into the confinement of gauge theories[5,6], explorations of quantum gravity[7], and advancements in the understanding of quantum entanglement[8,9]. An important instance to study defects is in the context of conformal field theory (CFT)[10,11], where one considers the situation of deforming a CFT with interactions living on a sub-dimensional defect. The defect may trigger an RG flow towards a non-trivial infrared (IR) fixed point, which can still have an emergent conformal symmetry defined on the space-time dimensions of the defect[12–17]. The theory describing such a conformal defect is called a defect CFT (dCFT) (see refs. 18,19 for recent discussions). Understanding dCFTs is an important step in comprehending CFTs in nature, as most experimental realizations of CFTs necessarily accompany defects (and boundaries). Moreover, dCFTs have a non-trivial interplay

with the bulk CFTs, and knowledge of the former will advance the understanding of the latter. For example, the two-point correlators of bulk operators in dCFT constrain and encode the conformal data of the bulk CFT[20], similar to the well-known story of four-point correlators of a bulk CFT.

dCFTs are typically richer and more intricate than their bulk CFT counterparts. On one hand, for a given bulk CFT, there exist multiple (even potentially infinite) distinct dCFTs, and their classification remains an open challenge. On the other hand, breaking of the full conformal symmetry group into a subgroup renders the study of dCFTs more challenging, as the space-time conformal symmetry becomes less restrictive, making modern approaches like the conformal bootstrap program[21] less powerful[20,22–25]. Notably, most of the well-established results concerning dCFTs are confined to 2D CFTs, including the seminal results on the boundary operator contents[13] and RG flow[26], thanks to the special integrability property of 2D CFTs. In comparison, higher-dimensional CFTs pose greater difficulties, and the knowledge of dCFTs in dimensions beyond two is rather limited.

[1]Department of Physics, School of Science, Westlake University, Hangzhou 310030, PR China. [2]Institute of Natural Sciences, Westlake Institute for Advanced Study, 18 Shilongshan Road, Hangzhou 310024, PR China. [3]Perimeter Institute for Theoretical Physics, Waterloo, ON N2L 2Y5, Canada. ✉ e-mail: yhe@perimeterinstitute.ca; zhuwei@westlake.edu.cn

Current studies of dCFTs mainly revolve around perturbative RG computations[27–34] and Monte Carlo simulations of lattice models[18,35–37]. An important progress made recently is the non-perturbative proof of RG monotonic g-theorem in 3D and higher dimensions[38,39], generalizing the original result in 2D[26,40,41].

In the context of dCFTs, many important questions remain to be answered, ranging from basic inquiries such as the existence of conformal defect fixed points to more advanced queries concerning the infrared properties of dCFTs, including their conformal data such as critical exponents. The central aim of this paper is to develop an efficient tool for the non-perturbative analysis of 3D dCFTs. Specifically, we extend the success of the recently proposed fuzzy sphere regularization[42] from bulk CFTs[42–45] to the realm of dCFTs. As a concrete example, we explore the properties of the 3D Ising CFT in the presence of a magnetic line defect[32–36,46–49]. We directly demonstrate that this line defect indeed flows to an attractive conformal fixed point, and we identify 6 low-lying defect primary operators with their scaling dimensions extracted through the state-operator correspondence. Furthermore, we study the one-point bulk primary correlators and the two-point bulk-defect correlators, both of which are fixed by conformal invariance, up to a set of operator product expansion (OPE) coefficients. As far as we know, most of conformal data of dCFT reported here have never been studied before. In this context, our paper not only presents a comprehensive set of results concerning the magnetic line defect in the 3D Ising CFT, but also lays the foundation for further exploration of 3D dCFTs using the fuzzy sphere regularization technique.

## Results

### Conformal defect and radial quantization

We consider a 3D CFT deformed by a $p$-dimensional defect, described by the Hamiltonian

$$H_{CFT} + h \int d^p r \mathcal{O}(r). \tag{1}$$

Examples include the line defect ($p = 1$, see Fig. 1a) and the plane defect ($p = 2$). If the defect is not screened in the IR, the system will flow into a non-trivial fixed point that breaks the original conformal symmetry $SO(4, 1)$ of $H_{CFT}$. Furthermore, if the non-trivial fixed point is still conformal, such a defect is called a conformal defect described by a dCFT. For such a dCFT, the original conformal group is broken down to a smaller subgroup $SO(p + 1, 1) \times SO(3 - p)$[17–19], where $SO(p + 1, 1)$ is the conformal symmetry of the defect, and $SO(3 - p)$ is the rotation symmetry around the defect that acts as a global symmetry on the defect.

A dCFT possesses a richer structure compared to its bulk counterpart. Firstly, there is a set of operators living on the defect, forming representations of the defect conformal group $SO(p + 1, 1)$. Furthermore, there are non-trivial correlators between bulk operators and defect operators. (Hereafter, we follow the usual convention and denote the defect operator with a hat $\hat{O}$, while the bulk operator is represented as $O$ without a hat.) The simplest example is that the bulk primary operator gets a non-vanishing one-point correlator, which is in sharp contrast to the bulk CFT[17–19]:

$$\langle O_1(x) \rangle = \frac{a_{O_1}}{|x_\perp|^{\Delta_1}}. \tag{2}$$

Here, $|x_\perp|$ is the perpendicular distance from the bulk operator to the defect, $\Delta_1$ is the scaling dimension of $O_1$, and $a_{O_1}$ is an operator-dependent universal number (we consider the case of $O_1$ to be a Lorentz scalar). Moreover, we can consider a bulk-defect two-point (scalar-scalar) correlator defined as[17–19]:

$$\langle O_1(x)\hat{O}_2(0) \rangle = \frac{b_{O_1\hat{O}_2}}{|x_\perp|^{\Delta_1 - \hat{\Delta}_2}|x|^{2\hat{\Delta}_2}}, \tag{3}$$

where $b_{O_1\hat{O}_2}$ is the bulk-defect OPE coefficient. Interestingly, the bulk two-point correlator already becomes non-trivial, and its functional form cannot be completely fixed by the conformal symmetry.

Similar to the bulk CFT, we consider the radial quantization of a dCFT. Specifically, we first foliate the Euclidean space $\mathbb{R}^3$ using spheres $S^2$ with their origins situated on the defect, as illustrated in Fig. 1a. Next, we can perform a Weyl transformation to map $\mathbb{R}^3$ to a cylinder $S^2 \times \mathbb{R}$, and the $p$-dimensional defect transforms into a defect intersecting the cylinder. For instance, as shown in Fig. 1, the Weyl transformation maps a line defect ($p = 1$) in $\mathbb{R}^3$ to $0 + 1$D point impurities located at the north and south poles of the sphere $S^2$, forming two continuous line cuts along the time direction from $\tau = -\infty$ to $\tau = \infty$. Similarly, a plane defect ($p = 2$) in $\mathbb{R}^3$ will be mapped to a $1 + 1$D defect with its spatial component located on the equator of the sphere $S^2$.

Akin to the state-operator correspondence in bulk CFT[50,51], we have a one-to-one correspondence between the defect operators and the eigenstates of the dCFT quantum Hamiltonian on $S^2 \times \mathbb{R}$, where energy gaps of these states are proportional to the scaling dimensions $\hat{\Delta}_n$ of the defect operators:

$$E_n - E_0 = \frac{v}{R}\hat{\Delta}_n. \tag{4}$$

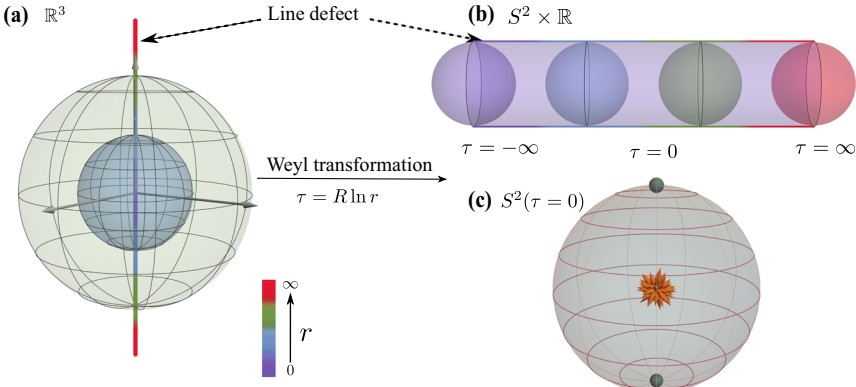

**(a)** $\mathbb{R}^3$    Line defect    **(b)** $S^2 \times \mathbb{R}$

Weyl transformation
$\tau = R \ln r$

$\tau = -\infty$    $\tau = 0$    $\tau = \infty$

**(c)** $S^2(\tau = 0)$

$\infty$

$r$

$0$

**Fig. 1 | Schematic plot of the defect in 3D.** Through a Weyl transformation, **a** Euclidean flat space-time $\mathbb{R}^3$ is mapped to (**b**) the cylinder manifold $S^2 \times \mathbb{R}$. The line defect before and after the Weyl transformation are shown by the colored line. **c** The $0 + 1$-D impurities (cyan point) located at the north and south pole on two-dimensional sphere $S^2$ in the radial quantization, where the flux at the center represents the magnetic monopole defined in the fuzzy sphere model.

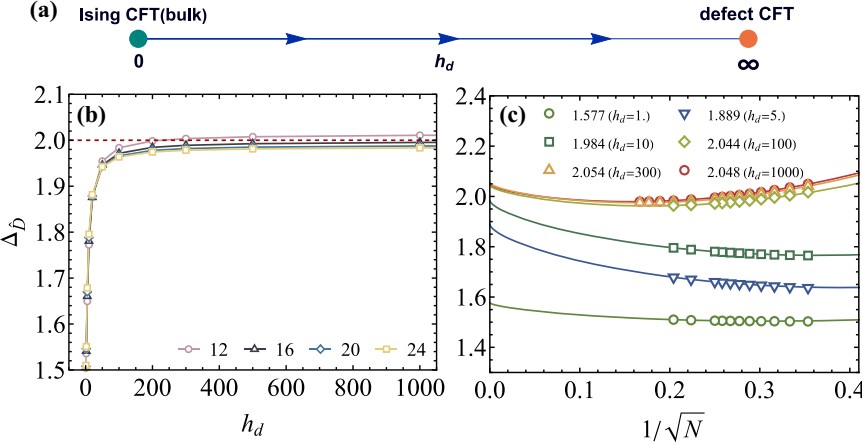

**Fig. 2 | Defect induced attractive fixed point. a** Schematic plot of the RG diagram. **b** The scaling dimension of displacement operator $\Delta_{\hat{D}}$ as a function of defect strength $h_d$. Different colored symbols represent the results based on various system sizes. **c** Finite-size extrapolation of $\Delta_{\hat{D}}$ for various $h_d$ (see Supplementary

Note 2 and 3 in Supplementary Material). A sufficient large $h_d$ gives almost identical $\Delta_{\hat{D}} \approx 2$, supporting an attractive RG fixed point at $h_d = \infty$. Different colored symbols represent the results based on various defect strength $h_d$.

Here, $E_0$ denotes the ground state energy of the defect Hamiltonian, $R$ represents the sphere radius, and $v$ is a model-dependent non-universal velocity that corresponds to the arbitrary normalization of the Hamiltonian. Notably, this velocity $v$ is identical to the velocity of the bulk CFT Hamiltonian (further discussions see Supplementary Note 2 in Supplementary Material).

The state-operator correspondence offers distinct advantages for studying CFTs. Firstly, it provides direct access to information regarding whether the conformal symmetry emerges in the IR. Secondly, it enables an efficient extraction of various conformal data, such as scaling dimensions and OPE coefficients of primaries. The key step involves studying a quantum Hamiltonian on the sphere geometry. However, for 3D CFTs, this was challenging as no regular lattice could fit $S^2$. Recently, this fundamental obstacle was overcome through a scheme called "fuzzy sphere regularization"[42], and its superior capabilities have been convincingly demonstrated[42–45]. Below we discuss how to adapt the fuzzy sphere regularization scheme to solve dCFTs. We will focus on the case of magnetic line defect of the 3D Ising CFT, but the generalizations to other cases should be straightforward.

**Magnetic line defect on the fuzzy sphere**

The fuzzy sphere regularization[42] considers a quantum mechanical model describing fermions moving on a sphere with a $4\pi s$ magnetic monopole at the center. The model is generically described by a Hamiltonian $H = H_{kin} + H_{int}$, where $H_{kin}$ represents the kinetic energy of fermions, and its eigenstates form quantized Landau levels described by the monopole Harmonics $Y_{n+s,m}^{(s)}(\theta,\varphi)$[52]. Here, $n = 0, 1, \cdots$ denotes the Landau level index, and $(\theta,\varphi)$ are the spherical coordinates. We consider the limit where $H_{kin}$ is much larger than the interaction $H_{int}$, allowing us to project the system onto the lowest Landau level (i.e. $n = 0$), resulting in a fuzzy sphere[53].

The 3D Ising transition on the fuzzy sphere can be realized by two-flavor fermions $\boldsymbol{\psi}^\dagger = (\psi_\uparrow^\dagger, \psi_\downarrow^\dagger)$ with interactions that mimic a 2+1D transverse Ising model on the sphere,

$$
\begin{aligned}
H_0 &= \int R^4 d\Omega_a d\Omega_b\, U(\Omega_{ab})(n^0(\Omega_a)n^0(\Omega_b) \\
&\quad -n^z(\Omega_a)n^z(\Omega_b)) - h \int R^2 d\Omega\, n^x(\Omega).
\end{aligned}
\tag{5}
$$

Here we are using the spherical coordinate $\Omega = (\theta,\phi)$ and $R$ is the sphere radius. The density operators are defined as $n^\alpha(\Omega) = \boldsymbol{\psi}^\dagger(\Omega) \sigma^\alpha \boldsymbol{\psi}(\Omega)$, where $\sigma^{x,y,z}$ are the Pauli matrices and $\sigma^0$ is the identity matrix. $U(\Omega_{ab})$ encodes the Ising density-density interaction as $U(\Omega_{ab}) = \frac{g_0}{R^2}\delta(\Omega_{ab}) + \frac{g_1}{R^4}\nabla^2\delta(\Omega_{ab})$. One can tune the transverse field $h$

to realize a phase transition which falls into the 2+1D Ising universality class[42]. In the following, we set $U(\Omega_{ab})$ and $h$ the same as the bulk Ising CFT that has been identified in[42].

To study the magnetic line defect of 3D Ising CFT, we add 0 + 1D point-like magnetic impurities located at sphere's north and south pole, modeled by a Hamiltonian term,

$$
H_d = 2\pi h_d(n^z(\theta = 0, \varphi = 0) + n^z(\theta = \pi, \varphi = 0)),
\tag{6}
$$

where $h_d$ controls the strength of the magnetic impurities. This type of defect can be artificially realized in experiments[54,55]. Crucially, the defect term $H_d$ breaks the Ising $\mathbb{Z}_2$ symmetry, causing the $\sigma$ field (of the 3D Ising CFT) to be turned on at the defect. This $\sigma$ deformation is relevant on the line defect ($\Delta_\sigma \approx 0.518 < 1$[21]), driving the system to flow to a nontrivial fixed point, conjectured to be a conformal defect. This fixed point is expected to be an attractive fixed point[32–34], implying that regardless of the strength of $h_d$, the defect will flow to the same conformal defect fixed point (see Fig. 2a). Next we will provide compelling numerical evidence to support this conjecture.

**Emergent conformal symmetry and operator spectrum**

The energy spectrum of the defect Hamiltonian ($H_0 + H_d$) is expected to be proportional to the defect operators' scaling dimensions, up to a non-universal velocity in Eq. (4). Here we determine the velocity using the bulk CFT Hamiltonian ($H_0$) by setting the $\sigma$ state to have $\Delta_\sigma = 0.51814$[21]. The defect term $H_d$ breaks the sphere rotation $SO(3)$ down to $SO(2)$, so each eigenstate has a well defined $SO(2)$ quantum number $L_z$. Akin to the stress tensor of the bulk CFT, there exists a special primary operator in dCFT due to the broken of translation symmetry, dubbed the displacement operator $\hat{D}$[17–19], which has $L_z = \pm 1$ and a protected scaling dimension $\Delta_{\hat{D}} = 2$. Fig. 2b, c depicts $\Delta_{\hat{D}}$ via the state-operator correspondence (Eq. (4)) for various defect strength $h_d$ and system sizes. It clearly shows that the obtained $\Delta_{\hat{D}}$ are very close to 2, for different defect strengths $h_d$, which indicates an attractive conformal fixed point at $h_d = \infty$ (see Supplementary Note 3 and 6 in Supplementary Material). In what follows, we present the representative results for $h_d = 300$ and we ensure the conclusions are insensitive to the choice of $h_d$.

We further establish the emergent conformal symmetry by confirming that the excitation spectra form representations of $SO(2, 1)$. The generators of $SO(2, 1)$ are the dilation $D$, translation $P$, and special conformal transformation $K$. It is important to note that $P$ and $K$ do not have any Lorentz index due to the triviality of the Lorentz symmetry,

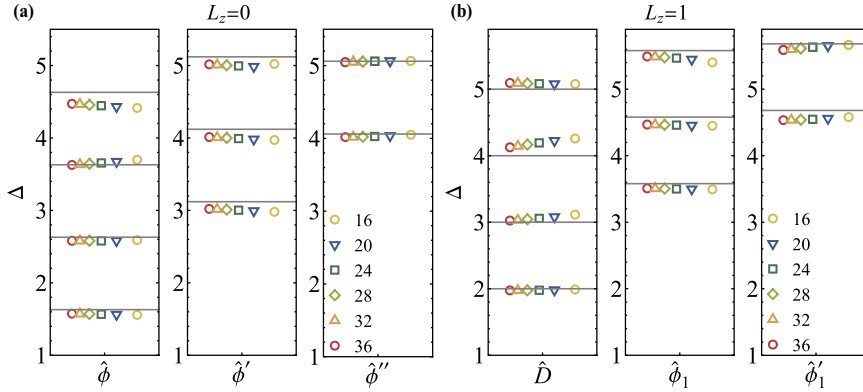

**Fig. 3 | Conformal tower of defect primaries.** Defect primary fields and their descendants with global symmetry (**a**) $L_z = 0$ and (**b**) $L_z = 1$. The gray horizon lines stand for extrapolated values for primaries and their integer-spaced descendants. Different colored symbols represent the results based on various system sizes. By increasing system size $N$ all of the scaling dimensions approach the theoretical values consistently, supporting an emergent conformal symmetry in the thermodynamic limit.

i.e., $SO(1)$. For each primary operator, we have descendants generated by the translation, $P^n\hat{O}$, whose scaling dimension is $\Delta_{P^n\hat{O}} = \Delta_{\hat{O}} + n$, and its $SO(2)$ quantum number $L_z$ remains unchanged. Figure 3 displays our numerical data of the low-lying energy spectrum, clearly exhibiting the emergent conformal symmetry, i.e., approximate integer spacing between each primary and its descendants. These observations firmly establish that the magnetic line defect of the 3D Ising CFT flows to a conformal defect with a conformal symmetry of $SO(2,1)$.

From our numerical data, we can identify five low-lying defect primary operators in addition to $\hat{D}$, as listed in Table 1. Notably, all these operators are found to be irrelevant (i.e., $\Delta > 1$), which is consistent with the observation of an attractive defect fixed point. Our lowest-lying operator $\hat{\phi}$ has $L_z = 0$ and $\Delta_{\hat{\phi}} \approx 1.63(6)$. This value is in good agreement with Monte Carlo simulations, e.g. $1.60(5)^{[36]}$, $1.52(6)^{[36]}$, and $1.40(3)^{[35]}$, as well as with the perturbative $\varepsilon$-expansion computation of $\sim 1.55(14)$ ref. 34. The second low-lying operator in the $L_z = 0$ sector has $\Delta_{\hat{\phi}'} = 3.12(10)$, which significantly deviates from the $\varepsilon$-expansion value of $\Delta \approx 4.33 + O(\varepsilon^2)$ (it was called $\hat{s}_+$ in[34]). This suggests a large subleading correction in the $\varepsilon$-expansion. All other primary operators identified in our study have not been computed by any other methods. It is essential to mention that the scaling dimensions in Table 1 are obtained by the finite-size extrapolating (see details in Supplementary Note 2 in Supplementary Material), and the data at finite $N$ is already very close to the extrapolated value (The finite-size extrapolation improve the results by around 2%). One can also improve the accuracy by making use of conformal perturbation[56].

Additionally, to verify the physics of dCFT presented here is independent of the specific value $h_d$, we directly study the spectrum at $h_d = \infty$ (see Supplementary Note 6 in Supplementary Material). Comparing it with the results at $h_d = 300$, we found them to be consistent with each other. This result indicates that the large $h_d$ regime shares the same dCFT and also supports that the fixed point of dCFT indeed resides at $h_d = \infty$.

**Table 1 | Scaling dimensions of primary operators in the magnetic line defect of 3D Ising CFT, determined through the state-operator correspondence on the fuzzy sphere**

| $L_z = 0$ | | | $L_z = 1$ | | |
|---|---|---|---|---|---|
| $\hat{\phi}$ | $\hat{\phi}'$ | $\hat{\phi}''$ | $\hat{D}$ | $\hat{\phi}_1$ | $\hat{\phi}_1'$ |
| 1.63(6) | 3.12(10) | 4.06(18) | 2.05(7) | 3.58(7) | 4.64(14) |

Please see a detailed analysis of errors and finite-size extrapolation in Supplementary Note 2 and 3 in Supplementary Material.

## Correlators and OPE coefficients

Using Weyl transformation, we can map the bulk-defect correlators in Eq. (2), Eq. (3) in $\mathbb{R}^3$ to the correlators on cylinder $S^2 \times \mathbb{R}$ (see Supplementary Note 1 in Supplementary Material),

$$G_{O_1\hat{O}_2} \equiv \frac{\langle\hat{1}|O_1(\tau=0,\theta)|\hat{O}_2\rangle}{\langle\hat{1}|O_1(\tau=0,\theta)|\hat{1}\rangle} = \frac{b_{O_1\hat{O}_2}}{(\sin\theta)^{\Delta_1 - \hat{\Delta}_2}}. \quad (7)$$

The bulk operator $O_1$ is positioned at a point that has an angle $\theta$ with respect to the north pole. In the denominator, we use the states of the bulk CFT, while in the numerator, we use the states of the dCFT. The one-point bulk correlator corresponds to taking $|\hat{O}_2\rangle$ to be the ground state of the defect, i.e., $|\hat{1}\rangle$.

In the fuzzy sphere model, we can use the spin operators $n^z$ and $n^x$ to approximate the bulk CFT primary operators $\sigma$ and $\epsilon^{[43,44]}$. For example, the correlator between the bulk primary $\sigma$ and a defect primary operator $\hat{O}_2$ is computed by,

$$G_{\sigma\hat{O}_2} \equiv \frac{\langle\hat{1}|n^z(\tau=0,\theta)|\hat{O}_2\rangle}{\langle\hat{1}|n^z(\tau=0,\theta)|\sigma\rangle} = \frac{b_{\sigma\hat{O}_2}}{(\sin\theta)^{\Delta_\sigma - \hat{\Delta}_2}} + O(N^{-1/2}). \quad (8)$$

Here, $\Delta_\sigma \approx 0.518149$, and the first-order correction $O(N^{-1/2})$ comes from the descendant operator $\partial_\mu\sigma$ contained in $n^z$. Figure 4 illustrates the one-point bulk correlator $G_\sigma(\theta)$ and bulk-defect correlator $G_{\sigma\hat{\phi}}(\theta)$ for different system sizes $N = 12$–$36$. Both correlators agree perfectly with the CFT prediction Eq.(7), except for the small $\theta$ regime. It is worth noting that the one-point correlator $G_\sigma(\theta)$ is divergent at $\theta = 0, \pi$ and reaches a minimum at $\theta = \pi/2$. In contrast, the bulk-defect correlator $G_{\sigma\hat{\phi}}(\theta)$ has an opposite behavior (because $\Delta_\sigma - \Delta_{\hat{\phi}} < 0$); it vanishes at $\theta = 0, \pi$ and reaches a maximum at $\theta = \pi/2$. These behaviors are nicely reproduced in our data, which is highly nontrivial because computationally the only difference for the two correlators is the choice of $|\hat{O}_2\rangle$ in Eq. (8).

We can further extract the bulk-defect OPE coefficients from $G_{O_1\hat{O}_2}(\theta = \pi/2) = b_{O_1\hat{O}_2}$, and the results are summarized in Table 2. None of these OPE coefficients was computed non-perturbatively before. There are perturbative computations for $a_\sigma$ and $a_\epsilon^{[34]}$ from $\varepsilon$ expansion, giving $a_\sigma^2 \approx 3.476 + O(\varepsilon^2)$ (i.e. $a_\sigma \approx 1.86$) and $a_\epsilon \approx 1.83 + O(\varepsilon^2)$. Our estimates are $a_\sigma = 1.37(1)$ and $a_\epsilon = 1.31(19)$, it will be interesting to compute higher order corrections in the $\varepsilon$-expansion. Moreover, using the Ward identity of any bulk operator $(O)^{[19]}$,

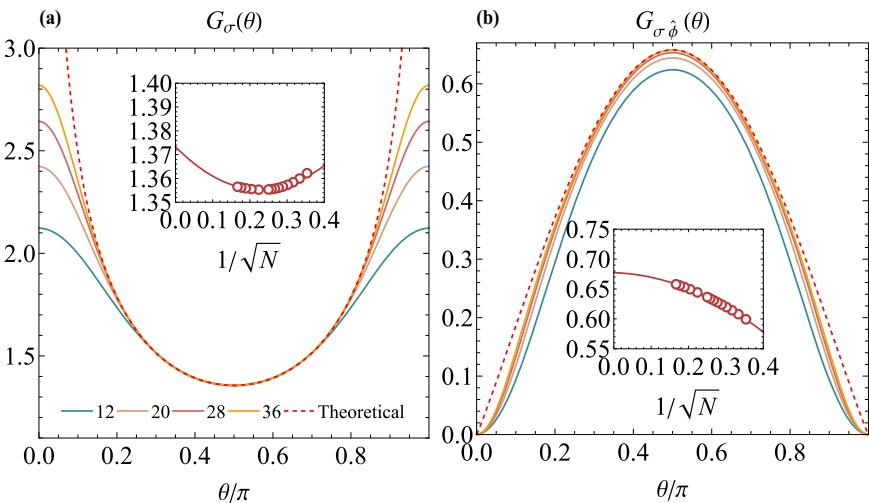

**Fig. 4 | Correlators involving defect.** The angle dependence of (**a**) correlator $G_\sigma(\theta)$ and (**b**) $G_{\sigma\hat\phi}(\theta)$, for system sizes ranging from $N = 12-36$. The dashed lines correspond to theoretical correlator in Eq. (7) with $b_{\sigma\hat O_2}$ and $\Delta_{\hat O_2}$ from the $N = 36$ curves.

The insets are finite-size scaling analysis by setting $\theta = \pi/2$, respectively giving one-point OPE coefficient $a_\sigma \approx 1.37(1)$ and bulk-defect OPE coefficient $b_{\sigma\hat\phi} \approx 0.68(1)$.

we can extract Zamolodchikov norm

$$\sqrt{2C_{\hat D}} = \frac{2}{\pi}\frac{\Delta_O G_O(\theta = \pi/2)}{G_{O\hat D}(\theta = \pi/2)}. \tag{9}$$

The estimates using $\sigma$ and $\epsilon$ gives $C_{\hat D} = 0.27(1)$ and $C_{\hat D} = 0.30(8)$, respectively.

## Discussion

We have outlined a systematic procedure to study defect conformal field theory (dCFT) using the recently proposed fuzzy sphere regularization scheme. As a concrete application, we investigated the magnetic line defect of 3D Ising CFT and provided clear evidence that it flows to a conformal defect. Crucially, we accurately computed a number of conformal data of this dCFT, including defect primaries' scaling dimensions and bulk-defect OPE coefficients. As far as we know, most of the conformal data of dCFT reported here have never been studied in a microscopic model before, thus this conformal information paves the way for exploring the rich physics in 3D Ising CFTs.

Looking forward, the current setup can be readily applied to the study of various types of defects in distinct 3D CFTs, potentially resolving numerous open questions and offering insights into defects in CFTs. For example, the plane defect ($p = 2$ in Eq. (1)), which may resemble surface critical phenomena, is interesting to investigate. It is also highly desired to study 3D dCFTs in a broad universality class (e.g. Wilson–Fisher $O(N)$ critical point). Moreover, the results of current work provide a necessary input in the study of infrared data for the dCFT within the numerical conformal bootstrap[20-22]. Taking it further would potentially be very interesting to study the dCFT in holography and string theory.

**Table 2 | Bulk-to-defect OPE coefficients magnetic line defect of 3D Ising CFT**

| $a_\sigma$ | $b_{\sigma\hat\phi}$ | $a_\epsilon$ | $b_{\epsilon\hat\phi}$ | $C_{\hat D}$ by $\sigma$ | $C_{\hat D}$ by $\epsilon$ |
|---|---|---|---|---|---|
| 1.37(1) | 0.68(1) | 1.31(19) | 1.63(4) | 0.27(1) | 0.30(8) |

$C_{\hat D}$ is computed by Eq. (9) using $\sigma$ and $\epsilon$.

## Methods

The model $H_0 + H_d$ for the magnetic line defect of 3D Ising CFT is a continuous model with fully local interaction in the spatial space. In practice, we consider the second quantization form of this model by the projecting $H_0 + H_d$ to the lowest Landau level (fuzzy sphere), using $\psi_a(\Omega) = \frac{1}{\sqrt{N}}\sum_{m=-s}^s c_{m,a}Y_{s,m}^{(s)}(\Omega)$ (we are using a slightly different convention compared to ref. 42). Here $N = 2s + 1$ playing the role of system size $N \sim R^2$, and we simply replace $R^2$ with $N$ during the projection. This lowest Landau level projection leads to a second quantized Hamiltonian defined by fermionic operators $c_{m,a}$, and similar models have been extensively studied in the context of the quantum Hall effect[57]. Numerically, this model can be simulated using various techniques such as exact diagonalization and density matrix renormalization group (DMRG)[58,59]. We perform DMRG calculations with bond dimensions up to $D = 5000$, and for the largest system size $N = 36$, the maximum truncation errors for the ground state and the tenth excited state are $1.37 \times 10^{-9}$ and $1.96 \times 10^{-8}$, respectively. We explicitly impose two $U(1)$ symmetries, i.e., fermion number and $SO(2)$ angular momentum.

## Data availability

All data are included in this published article and Supplementary Information files.

## Code availability

The codes used to generate data and plots are available from the corresponding author upon request. The DMRG data are generated using the software "ITensor 3(C++ Version)"[60].

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

## Acknowledgements

We thank Davide Gaiotto for stimulating discussions that initiated this project. L.D.H. and W.Z. were supported by National Natural Science Foundation of China (No. 92165102, 11974288) and National key R&D program (No. 2022YFA1402204). Research at Perimeter Institute is supported in part by the Government of Canada through the Department of Innovation, Science and Industry Canada and by the Province of Ontario through the Ministry of Colleges and Universities. YCH thanks the hospitality of Bootstrap 2023 at ICTP South American Institute for Fundamental Research, where part of this project was done.

## Author contributions

W.Z. and Y.-C.H. initiated the project, L.D.H. performed the simulations. All authors contributed equally to the analysis of the data and writing of the manuscript.

## Competing interests

The authors declare no competing interests.
