## [Peer Review File · Nature Communications]

REVIEWER COMMENTS

Reviewer #1 (Remarks to the Author):

Defect CFTs (dCFTs) describe the IR dynamics of impurities at quantum critical points and are therefore of great interest. However, despite some recent theoretical advances, most of our understanding of dCFTs is based on perturbative calculations, e.g. within the epsilon expansion or at large N . This manuscript shows that the recently proposed fuzzy sphere regularization can be applied to studying dCFTs at the non-perturbative level.

The manuscript focuses on the magnetic line defect in the Ising model. This is perhaps the simplest example of a defect CFT (dCFT), and it has been the subject of several theoretical works, especially within the last few years. The authors convincingly show that the magnetic line defect exhibits conformal symmetry in the infrared, and measure several critical exponents and OPE coefficients. The method's validity is demonstrated by the agreement between the manuscripts' results and the few solid expectations available in the literature, including the scaling dimension of the lightest primary operator (formerly measured via Monte-Carlo simulations) and the Ward identity for the displacement operator. However, most of the results obtained in the manuscript are new and illustrate the power of the method (over, say, conventional Monte-Carlo simulations).

Because of the reasons mentioned above, I believe that this work provides an important contribution and is suitable for publication in Nature Communications. Before that, I would like the authors to consider the following minor observations:

- In Fig. 3., how is the grey line for the primary operators obtained? The data points seem to lie below it.
- In the supplemental material, I suspect that eq.s S6 and S10 should include a term $\sim R^{2(\Delta_{\phi}-1)}$, that arises at second order in conformal perturbation theory from the leading irrelevant deformation; unless this term vanishes for some reason, the resulting contribution is more important than the one proportional to c in eq. S6 and must be included.
- Relatedly, shouldn't one also include the effect of the leading bulk irrelevant deformation in the conformal perturbation theory formulas for the dCFT data? Or has that operator been tuned away already?
- I found some typos: 1) in the abstract "...dissect the defects of 3D CFTs..." 2) In the conclusion, the last line "holographic" \rightarrow holography

Reviewer #2 (Remarks to the Author):

The authors investigate the universality class of the critical tridimensional Ising model in the presence of a (magnetic) defect line using Fuzzy sphere regularization. The approach employed here generalizes the methods considered in PRX X 13, 021009 (2023), where the corresponding homogeneous model was studied. The key idea is to focus on a 2-dimensional quantum model of interacting fermions on a sphere, whose single-particle modes are restricted to the Lowest Landau Level (LLL) arising in the presence of a magnetic monopole. A Z_2 symmetry is present in the homogeneous theory, which is explicitly broken by a localized defect corresponding to the line defect in the associated field theory.

The main advantage of the approach above, compared, for example, with lattice regularizations, is the "natural" realization of some symmetries in the corresponding Conformal Field Theory (CFT), such as rotations around the line defect. Therefore, the expectation, supported somewhat by numerical evidence, is a faster convergence of the data obtained at finite dimension (of the LLL) toward the universal value of the CFT.

The obtained data appear reasonable. While the crucial assumptions rely on a conjecture—the emergence of the Ising universality class—the extrapolated values for the scaling dimension seem compatible with the theory (e.g., for the displacement operator and the tower of descendants) and with available Monte Carlo data. Moreover, a comparison with older predictions obtained via epsilon expansion is done, though it is not conclusive due to the non-accuracy of those predictions (related to their "perturbative" validity).

The work suggests many possible directions. For example, other universality classes might be realized as well, such as the $O(N)$ models or Potts model in the presence of impurities that partially break the symmetry. Also, non-trivial relations between CFTs, Quantum Hall physics, and 2D CFTs are left open.

I think the manuscript satisfies the criteria for acceptance, especially regarding the originality of the methods to address long-standing problems in statistical physics. I do not have any major observations, just a small comment:

The data are obtained as a function of h_d , and until $h_d = 1000$ there are indications of convergence. I would say that two possible scenarios can occur in principle:

1. The model is well-behaved for $h_d \rightarrow \infty$. In this regime, the spectrum should be split into "sectors" associated with the eigenvalues of the defect perturbation, and whose separation in energy

is of the order of h_d . If this occurs, one might, in principle, project the model onto the lightest sector, study the eigenvalues there, reaching higher precision data compared to the values at finite h_d .

2. The model "breaks down" if h_d is too large. In particular, a critical value of h_d , which depends on the "size of the system" (N), is present, and the universality class of the defect CFT is not reached anymore. If this is the case, a dramatic change in behavior might be observed in the data at very large h_d , and, in principle, one should be careful about the relationship between the size of the system and the perturbation parameter h_d .

Which scenario is conjectured by the authors? They might consider adding a small paragraph commenting on that to slightly improve the quality of the manuscript.

FIRST REFEREE

Comment: Defect CFTs (dCFTs) describe the IR dynamics of impurities at quantum critical points and are therefore of great interest. However, despite some recent theoretical advances, most of our understanding of dCFTs is based on perturbative calculations, e.g. within the epsilon expansion or at large N . This manuscript shows that the recently proposed fuzzy sphere regularization can be applied to studying dCFTs at the non-perturbative level.

The manuscript focuses on the magnetic line defect in the Ising model. This is perhaps the simplest example of a defect CFT (dCFT), and it has been the subject of several theoretical works, especially within the last few years. The authors convincingly show that the magnetic line defect exhibits conformal symmetry in the infrared, and measure several critical exponents and OPE coefficients. The method's validity is demonstrated by the agreement between the manuscripts' results and the few solid expectations available in the literature, including the scaling dimension of the lightest primary operator (formerly measured via Monte-Carlo simulations) and the Ward identity for the displacement operator. However, most of the results obtained in the manuscript are new and illustrate the power of the method (over, say, conventional Monte-Carlo simulations).

Because of the reasons mentioned above, I believe that this work provides an important contribution and is suitable for publication in *Nature Communications*. Before that, I would like the authors to consider the following minor observations:

Reply: First of all, we thank the referee for the nice summary and recommendation.

Comment: In Fig. 3., how is the grey line for the primary operators obtained? The data points seem to lie below it.

Reply: For each primary field \hat{O} , the dashed line represents the extrapolated scaling dimensions $\Delta_{\hat{O}}$, based on finite-size scaling process using the formula in Eq. S10. For the descendant fields, the dashed lines are plotted using the scaling dimensions $\Delta_{\hat{O}} + n$ ($n = 1, 2, 3, \dots$) according to conformal symmetry. Especially, the displacement operator \hat{D} is expected to have theoretical value of scaling dimension $\Delta_{\hat{D}} = 2$, and dashed line is plotted using this exact value. We have now added a sentence in the caption to clarify it, "The grey horizon lines stand for extrapolated values for primaries and their integer-spaced

descendants.”

Comment: *In the supplemental material, I suspect that eq.s S6 and s10 should include a term $R^{-2(\Delta_\phi-1)}$, that arises at second order in conformal perturbation theory from the leading irrelevant deformation; unless this term vanishes for some reason, the resulting contribution is more important than the one proportional to c in eq. S6 and must be included.*

Reply: We thank the referee for raising this important question. In our previous manuscript, we only consider the first order perturbation and we fit the numerical data using the scaling function involving the operator $\hat{\phi}$ (with scaling dimension ~ 1.63) and $\hat{\phi}'$ (with scaling dimension ~ 3.12):

$$\Delta_{\hat{O}}(N) \approx \Delta_{\hat{O}} + \frac{b}{R^{\Delta_{\hat{\phi}}-1}} + \frac{c}{R^{\Delta_{\hat{\phi}'}-1}} + \text{higher order corrections.} \quad (1)$$

As the referee pointed out, if considering the second order perturbation, there is an additional contribution from the operator $\hat{\phi}$:

$$\Delta_{\hat{O}}(N) \approx \Delta_{\hat{O}} + \frac{b'}{R^{\Delta_{\hat{\phi}}-1}} + \frac{c'}{R^{2(\Delta_{\hat{\phi}}-1)}} + \text{higher order corrections.} \quad (2)$$

Firstly, we estimate that the first-order corrections are small, because the extrapolated scaling dimensions $\Delta_{\hat{O}}$ are quite close to the finite-size data $\Delta_{\hat{O}}(N)$ (please see Fig. S2 in the supplemental materials). Therefore, based on the results from the first order corrections, the contribution of the second order perturbation should be even smaller (say, the coefficient c' should be about one order smaller than b'). This is the main reason that we neglect the second order corrections in the scaling function.

Secondly, we can directly compare the finite-size extrapolation process using Eq. 1 and Eq. 2. As shown in Fig. 1, overall two scaling functions are very close to each other, and only a small difference is observed. For example, by employing Eq. 1 we obtain (for $h_d = 300$): $\Delta_{\hat{\phi}} \approx 1.625$ and $\Delta_{\hat{\phi}'} \approx 3.119$, while Eq. 2 gives $\Delta_{\hat{\phi}} \approx 1.617$ and $\Delta_{\hat{\phi}'} \approx 3.079$. Actually this small difference is even smaller than the error bar that we determined (please see the error analysis section). This consistency between Eq. 1 and Eq. 2 holds for most of the operators. (For the displacement operator \hat{D} , we observe a relatively large discrepancy $\sim 5\%$. This discrepancy is attributed to the overfitting using Eq. 2, since the fitted coefficient c' is unreasonably large.) Thus, in this paper, we utilize the scaling function from the first order perturbation Eq. 1 to do the numerical extrapolation. We have now included discussions about this in the supplementary materials of the revised version.

FIG. 1: Comparing the finite-size extrapolation using Eq. 1 (up to the first order perturbation) and Eq. 2 (based on the second order perturbation). The solid line represents the fitting process using Eq. 1, while the dashed line represents the results using Eq. 2. Here we set $h_d = 300$.

Comment: Relatedly, shouldn't one also include the effect of the leading bulk irrelevant deformation in the conformal perturbation theory formulas for the d CFT data? Or has that operator been tuned away already?

Reply: Yes, the referee is correct in stating that, in principle, the bulk irrelevant deformation should also affect the defect conformal data. Nevertheless, the leading bulk irrelevant operator perturbation (ϵ' with scaling dimension ~ 3.83) has been fine-tuned to be very small in our model by adjusting the ratio V_1/V_0 of the bulk Hamiltonian. At the optimal ratio $V_1/V_0 = 3.16$, the spectrum's dependence on the system size is notably weak, indicating that this leading irrelevant bulk perturbation is small. Similar to the previous question raised by the referee regarding second-order perturbation, it is theoretically correct to account for the effect of bulk perturbations. However, in practice, the changes are smaller than the error bars we have established. Moreover, incorporating additional terms in the extrapolation process could lead to overfitting.

Comment: I found some typos: 1) in the abstract dissect the defectS of 3D CFTs... 2) In the conclusion, the last line holographic \rightarrow holography

Reply: We greatly appreciate the referee for pointing out these typos. We have thor-

oroughly reviewed the entire manuscript and corrected the typos accordingly.

SECOND REFEREE

Comment: *The authors investigate the universality class of the critical tridimensional Ising model in the presence of a (magnetic) defect line using Fuzzy sphere regularization. The approach employed here generalizes the methods considered in PRX X 13, 021009 (2023), where the corresponding homogeneous model was studied. The key idea is to focus on a 2-dimensional quantum model of interacting fermions on a sphere, whose single-particle modes are restricted to the Lowest Landau Level (LLL) arising in the presence of a magnetic monopole. A Z_2 symmetry is present in the homogeneous theory, which is explicitly broken by a localized defect corresponding to the line defect in the associated field theory.*

The main advantage of the approach above, compared, for example, with lattice regularizations, is the "natural" realization of some symmetries in the corresponding Conformal Field Theory (CFT), such as rotations around the line defect. Therefore, the expectation, supported somewhat by numerical evidence, is a faster convergence of the data obtained at finite dimension (of the LLL) toward the universal value of the CFT.

The obtained data appear reasonable. While the crucial assumptions rely on a conjecture the emergence of the Ising universality class the extrapolated values for the scaling dimension seem compatible with the theory (e.g., for the displacement operator and the tower of descendants) and with available Monte Carlo data. Moreover, a comparison with older predictions obtained via epsilon expansion is done, though it is not conclusive due to the non-accuracy of those predictions (related to their "perturbative" validity).

The work suggests many possible directions. For example, other universality classes might be realized as well, such as the $O(N)$ models or Potts model in the presence of impurities that partially break the symmetry. Also, non-trivial relations between CFTs, Quantum Hall physics, and 2D CFTs are left open.

I think the manuscript satisfies the criteria for acceptance, especially regarding the originality of the methods to address long-standing problems in statistical physics. I do not have any major observations, just a small comment:

Reply: First of all, we thank the referee for a nice summary of our manuscript and his/her strong recommendation.

Comment: *The data are obtained as a function of h_d , and until $h_d = 1000$ there are indications of convergence. I would say that two possible scenarios can occur in principle:*

1. The model is well-behaved for $h_d \rightarrow \text{infinity}$. In this regime, the spectrum should be split into "sectors" associated with the eigenvalues of the defect perturbation, and whose separation in energy is of the order of h_d . If this occurs, one might, in principle, project the model onto the lightest sector, study the eigenvalues there, reaching higher precision data compared to the values at finite h_d .

2. The model "breaks down" if h_d is too large. In particular, a critical value of h_d , which depends on the "size of the system" (N), is present, and the universality class of the defect CFT is not reached anymore. If this is the case, a dramatic change in behavior might be observed in the data at very large h_d , and, in principle, one should be careful about the relationship between the size of the system and the perturbation parameter h_d .

Which scenario is conjectured by the authors? They might consider adding a small paragraph commenting on that to slightly improve the quality of the manuscript.

Reply: We thank the referee for this insightful question. First of all, we believe the answer to this question is the scenario 1 proposed by this referee, and we will explain it in detail as following.

Indeed, the defect term in the Hamiltonian only affects the $m = \pm s$ orbitals in orbital (angular momentum) space (see also arxiv.2401.00039).

$$\begin{aligned} H_d &= 2\pi h_d [n^z(\theta = 0, \varphi = 0) + n^z(\theta = \pi, \varphi = 0)] \\ &= \frac{1}{2} h_d \left(\hat{c}_s^\dagger \sigma^z \hat{c}_s + \hat{c}_{-s}^\dagger \sigma^z \hat{c}_{-s} \right). \end{aligned} \tag{3}$$

Thus, the $h_d \rightarrow \infty$ limit just fixes the $m = \pm s$ orbitals to \downarrow . So we directly projects the system into this sector, which improves computational efficiency. Furthermore, we have confirmed it numerically that, $h_d \rightarrow \infty$ limit is smoothly connected to finite but large h_d , as we will explain below.

After projecting the model onto the sector with spin fixed at the $m = s, -s$ orbitals, we recalculated the spectra for model sizes ranging from 8 to 36. It is evident that at $h_d = \infty$, the system still exhibits conformal symmetry similar to the finite h_d case(see Fig. 2), indicating it remains at the fixed point of dCFT. Furthermore, we compared the scaling dimensions of several lower primary fields with the finite $h_d = 300$ case mentioned in the main text. We found them to be consistent within the error bar(see Tab. I). This suggests that the choice of $h_d = 300$ in the main text is sufficiently large.

We also added a paragraph in the main text and a section in the supplementary material

FIG. 2: **Conformal tower of defect primaries in $h_d \rightarrow \infty$.** Defect primary fields and their descendants with global symmetry (a) $L_z = 0$ and (b) $L_z = 1$. The grey horizon lines stand for the theoretical expectation for descendants. Different colored symbols represent the results based on various system sizes. By increasing system size N all of scaling dimensions approach the theoretical values consistently, supporting an emergent conformal symmetry in the thermodynamic limit.

TABLE I: Scaling dimensions of primary operators in the magnetic line defect of 3D Ising CFT, determined through the state-operator correspondence on the fuzzy sphere.

	$L_z = 0$			$L_z = 1$		
	$\hat{\phi}$	$\hat{\phi}'$	$\hat{\phi}''$	\hat{D}	$\hat{\phi}_1$	$\hat{\phi}'_1$
$h_d = \infty$	1.63(4)	3.12(9)	4.04(3)	2.05(7)	3.58(7)	4.62(8)
$h_d = 300$	1.63(6)	3.12(10)	4.06(18)	2.05(7)	3.58(7)	4.64(14)

to discuss the infinite defect strength.